# Sensing and Responding to Hypersaline Conditions and the HOG Signal Transduction Pathway in Fungi Isolated from Hypersaline Environments: *Hortaea werneckii* and *Wallemia ichthyophaga*

**DOI:** 10.3390/jof7110988

**Published:** 2021-11-19

**Authors:** Ana Plemenitaš

**Affiliations:** Faculty of Medicine, Institute of Biochemistry and Molecular Biology, University of Ljubljana, Vrazov trg 2, 1000 Ljubljana, Slovenia; ana.plemenitas@mf.uni-lj.si

**Keywords:** halotolerant/halophilic fungi, *Hortaea werneckii*, Wallemia ichthyophaga, HOG, signal transduction pathway

## Abstract

Sensing and responding to changes in NaCl concentration in hypersaline environments is vital for cell survival. In this paper, we identified and characterized key components of the high-osmolarity glycerol (HOG) signal transduction pathway, which is crucial in sensing hypersaline conditions in the extremely halotolerant black yeast *Hortaea werneckii* and in the obligate halophilic fungus *Wallemia ichthyophaga*. Both organisms were isolated from solar salterns, their predominating ecological niche. The identified components included homologous proteins of both branches involved in sensing high osmolarity (SHO1 and SLN1) and the homologues of mitogen-activated protein kinase module (MAPKKK Ste11, MAPKK Pbs2, and MAPK Hog1). Functional complementation of the identified gene products in *S. cerevisiae* mutant strains revealed some of their functions. Structural protein analysis demonstrated important structural differences in the HOG pathway components between halotolerant/halophilic fungi isolated from solar salterns, salt-sensitive *S. cerevisiae*, the extremely salt-tolerant *H. werneckii*, and halophilic *W. ichthyophaga*. Known and novel gene targets of MAP kinase Hog1 were uncovered particularly in halotolerant *H. werneckii.* Molecular studies of many salt-responsive proteins confirm unique and novel mechanisms of adaptation to changes in salt concentration.

## 1. Introduction

Solar salterns and other hypersaline environments are extreme habitats that prevent the growth of most organisms except those that are well adapted to extremely high salt concentrations. Our laboratory has long been involved in studies of the molecular mechanisms of adaptations to extremely high NaCl concentrations. Focus is predominantly on two fungal model organisms isolated from solar salterns: the extremely halotolerant *Hortaea werneckii* and the obligate halophilic fungus *Wallemia ichthyophaga* [1,2,3,4,5,6]. While *H. werneckii* is unique in its adaptability to fluctuations in salt concentration and grows without NaCl as well as in the presence of up to 5 M NaCl, *W. ichthyophaga* is a true halophile that requires the presence of at least 10% of NaCl and also grows in the presence of up to 32% NaCl. Whole genome sequencing of both fungi, ion measurements, and studies of the adaptations of numerous proteins and membrane properties to high NaCl concentrations have revealed many new molecular mechanisms of adaptation. Some of these mechanisms are common to both fungi, but many are unique to halotolerant *H. werneckii* and halophilic *W. ichthyophaga*. Sensing changes in salt concentrations, which leads to appropriate responses regarding the expression of osmoresponsive genes, is vital for survival in environments with increased NaCl concentrations. 

We were therefore particularly interested in the high-osmolarity glycerol (HOG) signal transduction pathway, which is involved in sensing hyperosmolar conditions. Identifying components of the HOG pathway in *H. werneckii* and *W. ichthyophaga* and investigating their roles in the adaptation to hypersaline conditions demonstrated that both fungi are well adapted to such conditions. Furthermore, both fungi sense and respond to increased environmental salinity differently. In this review article, we summarize the key molecular adaptations of *H. werneckii* and *W. ichthyophaga*, focusing on the identification and structural properties of HOG pathway components. 

## 2. Molecular Adaptations of Halotolerant *H. werneckii*

The extremely halotolerant black yeast *H. werneckii* (Capnodiales, Dothideomycetidae, Dothideomycetes, Pezizomycotina) is described as an opportunistic pathogen and the primary etiological agent of tinea nigra, which can occur on the salty hands and feet of humans. It has been isolated from different hypersaline environments, with brine in solar salterns representing its primary ecological niche [2,7]. *H. werneckii* is an example of a very adaptable fungal species, as it can grow without salt as well as in environments with extremely high NaCl concentrations (up to 30%).

Molecular studies have identified general and also unique molecular mechanisms of adaptation that contribute to the extreme halotolerance of *H. werneckii* [1,4,5]. The most striking result came from the whole genome sequencing of *H. werneckii*, which was also the first published whole genome sequence of a fungal species isolated from an environment with an extremely high NaCl concentration. With almost 50 Mb, the *H. werneckii* genome is larger than the known genomes of other Capnodiales representatives, and, unlike in other species with large genomes, the large genome of *H. werneckii* is not due to large amounts of repeated DNA content. Analysis has demonstrated recent duplication of the *H. werneckii* genome, with multiple copies of halotolerance genes, in particular, genes that code for metal cation transporters [8,9]. The *H. werneckii* genome contains almost 16,000 genes of which 90% are duplicates with an average protein sequence divergence of 5%. While post-duplication events normally include large-scale gene deletions and reductions, the whole genome duplication appears stable in *H. werneckii*, as the genome was not reduced even after long-term growth in the presence of salt stress [9,10].

These whole genome data fit well with the results of molecular studies on individual genes. Most of the studied genes at the protein level are duplicated and code for two functionally identical proteins, often with a salt-dependent expression pattern. Such proteins are involved in general responses, as exemplified by the salt-dependent expression of two forms of glycerol phosphate dehydrogenase, which are involved in the synthesis of glycerol (the main compatible solute in *H. werneckii*) [11], fatty acid desaturases and elongases, which are involved in salt-dependent changes of membrane phospholipids [12,13,14], and especially metal cation transporters that are not only duplicated but exist as multiple copies in the *H. werneckii* genome [8]. The abundance of cation transporters in the *H. werneckii* genome indicates the importance of regulating potassium and sodium transport across the plasma membrane. Enrichment was not only observed for the transporters that maintain high intracellular K^+^/Na^+^ ratios and participate in Na^+^ efflux and K^+^ influx (e.g., *ENA*, *NHA*, and *TRK*) at high NaCl concentrations, but also for those involved in K^+^ efflux (*TOK*) and Na^+^ influx (*PHO*) at low NaCl concentrations [8,15,16,17]. This fits well with the lifestyle of *H. werneckii*, which thrives in environments with fluctuations in NaCl concentration, as maintenance of appropriate K^+^/Na^+^ ratios is crucial not only at high NaCl concentrations but also at low NaCl concentrations.

Two proteins with a specific biochemical signature of halophily were also identified in *H. werneckii*: hydroxymethylglutaryl CoA (HMG-CoA) reductase), the main regulatory enzyme in the mevalonate pathway [18,19], and 3′-phosphoadenosine-5′-phosphatase encoded by the HAL2 gene, a known target of salt toxicity in yeast [20]. NaCl-dependent regulation of HMG-CoA reductase in *H. werneckii* also appears to be linked to its halotolerant character. Both enzyme activity and protein levels in *H. werneckii* depend on environmental salinity, with high levels under both hypo- and hyper-saline conditions. Interestingly, we found that in *H. werneckii*, the biological consequence of HMG-CoA regulation relates to posttranslational protein modification by prenylation and not to sterol synthesis, as shown, for example, in *S. cerevisiae.* Thus, regulated expression and activity of HMG-CoA reductase can be considered an important biochemical signature of halophilism. In evolutionary terms, the maintenance of high HMG-CoA reductase levels in hypo- and hyper-saline environments may also reflect the physiological adaptation of halophilic fungi to metabolic demands under extreme conditions [19]. 

Another example of a protein with specific responses to changes in salinity is 3′-phosphoadenosine-5′-phosphatase Hal2. Two isoforms of the Hal2 protein appear to significantly contribute to the adaptation of *H. werneckii* to fluctuations in environmental salinity. With genetic and biochemical validation, the specific sequence motif of Hal2, which we call the META sequence, was identified to exert an effect on salt tolerance in the HwHal2B isoform as well as when introduced into the *SAL1* gene in *Arabidopsis thaliana* [20,21].

## 3. The HOG Signal Transduction Pathway in Halotolerant *H. werneckii*

The HOG signal transduction pathway is an example of a MAP kinase (mitogen-activated protein kinase) signal transduction pathway that senses and responds to hyperosmolar conditions and thus plays an important role in the adaptation to hypersaline conditions. It consists of the proteins of the sensory apparatus and the module of three kinases: MAPKKK (mitogen-activated protein kinase kinase kinase), MAPKK (mitogen-activated protein kinase kinase), and MAPK (mitogen-activated protein kinase); these are activated by sequential phosphorylation. This mechanism is best understood in the yeast *S. cerevisiae*. Hyperosmolar signals are sensed and transduced by two branches in *S. cerevisiae*: SHO1 and SLN1 that converge in the activation of MAPKK ScPbs2. In the SHO1 branch, the transmembrane ScSho1, membrane anchor ScOpy2, mucines ScMsb2 and ScHkr1, guanine exchange factor ScCdc24, and GTPase ScCdc42 are involved in the SHO1 sensory apparatus, which leads to the activation of ScSte20. Consequently, MAPKKK ScSte11 is activated by phosphorylation by the ScSte20 kinase. In the SLN1 branch, the proteins Sln1, Ypd1, and Ssk1 form the two-component phosphorelay system in which the sensor histidine kinase ScSln1 (located in the plasma membrane) is inactivated under hyperosmolar conditions. The ScYpd1 intermediate protein and the ScSsk1 response regulator then transduce the signal to the MAPKKKs ScSsk2/ScSsk22. The MAPKKKs ScSte11 and ScSsk2/ScSsk22 activate the MAPKK ScPbs2, which serves as a scaffold protein of the two upstream pathway branches and upon phosphorylation activates ScHog1 MAPK [22,23,24,25]. Under hyperosmotic conditions, ScHog1 phosphorylation is accompanied by an import of ScHog1 into the nucleus, which leads to the regulation of osmoresponsive genes. In addition, activated ScHog1 has targets in the cytosol and plasma membrane (18) that also play a role in osmoadaptation. The phosphorylation state of the MAPK Hog1 is transient and is controlled by various phosphoprotein phosphatases, such as Ser/ Thr phosphatase Ptc1 [26] and phosphotyrosine phosphatases Ptp2 and Ptp3 [27].

Homologous components to the *S. cerevisiae* HOG pathway have also been identified in other yeasts and fungi and are involved in responses to hyperosmolar conditions, virulence regulation, gut colonization, and animal and plant infection with pathogenic fungi. According to published data, it seems that two-component phosphorelay systems play important roles in sensing and subsequent HOG pathway activation in different fungi. For example, in the filamentous fungi *Aspergillus fumigatus* and *Aspergillus nidulans*, the HOG pathway is involved in osmotic stress responses [28,29], and the Sho1 protein is not involved in osmosensing in *A. nidulans* [28]. Likewise, the HOG pathway in the pathogenic fungus *Cryptococcus neoformans* is activated exclusively via the two-component phosphorelay system [30]. Histidine kinases were also identified as the main regulators of the HOG signal transduction pathway in *Magnaporthe oryzae* [31] and *Candida albicans* [32,33] 

Since the HOG pathway is highly conserved among eukaryotes, similar components to those in *S. cerevisiae* were expected in *H. werneckii*. Indeed, the genome of *H. werneckii* contains all the key components of the HOG signal transduction pathway. In line with the genome duplication in *H. werneckii*, we identified two copies of each gene that code for the components of the HOG pathway [8,9] (Figure 1). As there are no genetic tools available to allow manipulation of the *H. werneckii* genome, the putative roles of HOG pathway components identified in *H. werneckii* were studied by functional complementation in *S. cerevisiae* mutant strains [34,35,36]. 

### 3.1. The Sensory Apparatus of the HOG Pathway in H. werneckii

Although important differences were found in the sensory apparatus when compared to *S. cerevisiae*, it appears that, in contrast to other studied fungi, both the SLN1 and SHO1 branches play a role in sensing hypersaline conditions in *H. werneckii*, as we were able to identify components of both branches in *H. werneckii*. Interestingly, besides the *SLN1* gene, which codes for membrane-bound histidine kinase Sln1, the genome of *H. werneckii* contains two additional histidine kinase genes, *HwNIK1* and *HwHHK7*, which code for cytosolic histidine kinases Nik1 and Hhk7 [8,38]. While the role of membrane-bound Sln1 in sensing hyperosmolar conditions is well documented in *S. cerevisiae*, the role of the cytosolic histidine kinases, Nik1 and Hhk7, is less understood. Both were identified and proposed to be involved in sensing hyperosmolar conditions in pathogenic fungi [28,39]. 

We identified two isoforms of histidine kinase HwHHk7 in *H. werneckii*: HwHHk7A and B (Figure 1) [38]. Phylogenetic comparisons with other fungal histidine kinases (HK) and analyses of structural motifs confirmed that they belong to group 7 of fungal HKs. According to published data [40], histidine kinases Sln1 and HK 7 groups are positioned close together, and thus late separation from a common ancestor was proposed. However, these proteins differ in their intracellular localization: while histidine kinases from the Sln1 group are membrane-bound, histidine kinases from the HK7 group are soluble cytosolic proteins. It has been proposed that some of the histidine kinases other than yeast orthologs might transmit the signal through Ypd1; for example, in *A. nidulans*, the HOG pathway was reconstructed and revealed that the Sho1 branch is not involved in the osmoresponsive activation of the MAPK Hog1. It was speculated that this role could well be occupied by the HK 7 protein AnM7 [28,41]. 

To obtain insight into the role of the HwHHk7 isoforms in response to changes in salinity in halotolerant *H. werneckii*, we assessed the expression of both isoforms in response to changes in NaCl concentration. We found that the expression of the *HwHH7A* gene did not change significantly with changes in NaCl concentration. By contrast, transcription of the *HwHHK7B* gene was very salt-responsive. Moreover, we observed two types of responses: an early response with high induction of gene expression under hyposaline conditions and a late increased expression of the *HwHH7B* gene under hypersaline stress [38]. Based on these results, we proposed that in *H. werneckii*, the high induction of *HwHHK7B* gene expression as an early response to hyposaline stress could be the result of the specialized role of this HK in response to conditions of modest osmolarity, as has already been demonstrated for the Sln1 HK in *S. cerevisiae* [42]. *H. werneckii* is extremely adaptable to changes in NaCl concentration. Salt-dependent expression of the *HwHH7B* gene suggests that HwHhk7B plays an important role in sensing and adapting to the sudden changes that are common in solar salterns, natural habitats of *H. werneckii*.

Analysis of the *H. werneckii* genome also confirmed the presence of genes that code for proteins of the sensory apparatus of the SHO1 branch, membrane-bound HwSho1 A and B, HwOpy2 A and B, and HwMsb2 A and B. Structural and functional studies were performed on HwSho1 [34]. Comparison of amino-acid sequences of HwSho1 A and B in extremely halotolerant *H. werneckii* with those of salt-sensitive *S. cerevisiae, C. albicans,* and *A. fumigatus* and moderately halotolerant *Debaryomyces hansenii* showed highly conserved transmembrane and SH3 domains and a poorly conserved linker domain containing the Ste11-binding site. Despite differences in the Ste11-binding domain between *H. werneckii* and *S. cerevisiae* Sho1 proteins, both HwSho1 isoforms fully complemented the function of the native S. cerevisiae Sho1 protein when expressed in the *S. cerevisiae* strain with the deleted *SHO1* gene. Furthermore, they both activated the HOG pathway under conditions of osmotic stress. Moreover, we demonstrated that in S. cerevisiae, both HwSho1 proteins have characteristic subcellular localizations similar to the Sho1 protein [34] These data suggest that HwSho1 is involved in sensing hypersaline stress conditions in *H. werneckii.*

It appears that the sensory apparatus in the *H. werneckii* HOG pathway is much more complex when compared to *S. cerevisiae* and other plant and human pathogenic fungi. Based on our studies, we propose that the signaling to HwPbs2 in *H. werneckii* is conveyed through both branches: [43] the SHO1 branch, which is similar to that in *S. cerevisiae*, and [30] the SLN1 branch, which is similar to that in pathogenic fungi, such as *A. nidulans* and *C. albicans*, in which histidine kinases were proposed to be responsible for transmitting the signal to the MAP kinase module. We can speculate that the combination of both sensory systems in sensing changes in environmental salinity enables a more fine-tuned response to osmolarity in *H. werneckii*.

### 3.2. The MAP Kinase Module of the HOG Pathway in H. werneckii

Besides the genes of the sensory apparatus in the *H. werneckii* genome, gene duplication was also confirmed for downstream HOG pathway key components, including *HwSTE20*, *HwSTE50*, and genes that code for the three kinases of the MAP kinase module (*HwSTE11* A and B, *HwPBS2* A and B, and *HwHOG1* A and B). Further structural and functional studies of HwPbs2 and HwHog1 uncovered unique features of these kinases. Interestingly, in the case of the MAPKK HwPbs2, two gene copies of *HWPBS2* A and B are transcribed and translated into three different isoforms: while *HwPBS2A* is translated into one protein product (HwPbs2A), *HwPBS2*B is translated into two protein products (HwPbs2B1 and HwPbs2B2). Multiple isoforms were confirmed by Western blot analysis with specific antibodies raised against HwPbs2. It was found that they respond differently to changes in salt concentration. Their salt-responsive expression profiles suggest that HwPbs2a and HwPbs2b2 play roles in quick adaptation to severe hypersaline shock and that HwPbs2b1 plays a role in adaptation to moderate salt stress (Lenassi, Doctoral thesis, unpublished data).

The *H. werneckii* HOG signal transduction pathway includes two functionally redundant MAPK homologues, HwHog1A and HwHog1B. To better understand the function of the HOG pathway in *H. werneckii*, we observed the osmotic stress responsive phosphorylation pattern of HwHog1 and the effect of the Hog1 kinase inhibitor on cell viability and survival under hypersaline conditions. Activation of the HOG pathway in *H. werneckii* in response to increased osmolarity revealed that HwHog1A and HwHog1B are fully activated by constitutive phosphorylation with the upstream kinase HwPbs2 only at concentrations above 17% NaCl [44]. Thus, while phosphorylation is transient for most fungal species, this does not appear to be the case in extremely halotolerant *H. werneckii*. Likewise, the effect of HwHog1 kinase activity inhibition on restricting *H. werneckii* colony growth was only observed at osmolyte concentrations of ≥17%.

Studies on the *H. werneckii* transcriptome revealed many novel osmoresponsive genes, most of which were not shown before for *S. cerevisiae* or other fungi. Direct interaction with the MAP kinase HwHog 1 was demonstrated for more than one third of salt-responsive genes (Figure 1), with the majority of them upregulated in high-salinity-adapted *H. werneckii* cells [37] Genes associated with energy supply were highly represented among the upregulated genes in cells growing at 30% NaCl. An increase in energy production appears to be one of the fundamental adaptations that maintain ion homeostasis and osmotic equilibrium in a hyperosmotic environment. During long-term adaptive growth under extreme salinity, *H. werneckii* must maintain a high production of ATP that powers various transmembrane transporters [4,37,45]. In contrast to constitutive phosphorylation of HwHog1 in response to NaCl, the activation of osmoresponsive gene transcription was transient, which suggests that other factors besides the HwHog1 phosphorylation signal are involved in the effects of HwHog1 in *H. werneckii* [44].

We also showed that *H. werneckii* can discriminate between different osmolytes. NaCl induces continuous phosphorylation of HwHog1, whereas this phosphorylation is transient in response to KCl or sorbitol. Discrimination between osmolyte types was also confirmed at the level of osmoresponsive gene transcription. In *H. werneckii* cells, genes such as *HwSTL1* and *HwGPD1*, which are involved in glycerol transport and synthesis, showed early induction of transcription in response to NaCl stress, early or late induction in response to sorbitol stress, and no effects in response to KCl stress [44]. This osmolyte-specific response was also demonstrated in the expression of genes related to mitochondrial function. In hypersaline medium, an increased expression of genes involved in energy production and oxidative damage protection was observed, whereas adaptation to a non-ionic osmolyte resulted in a decrease in ATP synthesis and lipid peroxidation levels in mitochondria. This was also confirmed with a proteomic study of mitochondrial proteins, revealing a preferential accumulation of energy metabolism enzymes in hypersaline medium and an accumulation of protein chaperones in the non-ionic osmolyte [46].

In summary, the extremely halotolerant yeast-like fungus *H. wer**neckii*, isolated from hypersaline saltern water, can grow in both environments without salt as well as with extremely high salt concentrations (even up to saturated NaCl solutions). We assume that this extremely adaptable fungus has an efficient sensory system and molecular machinery that responds to changing conditions in the environment. As a response to severe changes in salinity in the environment, *H. werneckii* activates the HOG signaling pathway, resulting in the expression of many salt-responsive genes. The HOG signal transduction pathway is vital for the extreme osmotolerance of *H. werneckii* and regulates common and osmolyte-specific processes. Particularly, the HOG pathway seems to be specifically responsive to Na^+^, an ion typically present at high concentrations in the natural habitat of *H. werneckii*, solar eutrophic salterns.

## 4. Molecular Adaptations of the Halophilic Fungus *W. ichthyophaga*

*W. ichthyophaga* (Wallemiales, Wallemiomycetes) is a xerophilic filamentous fungi. Although xerotolerance is rare in Basidiomycota, *W. ichthyophaga* is the most halophilic fungus known. It requires a minimum of 10% *(w/v)* NaCl, with a growth optimum at 15–20% NaCl, but is metabolically active even at 32% NaCl [6] Strains of *W. ichthyophaga* have been isolated from hypersaline waters of solar salterns, bitterns, and salted meat [47,48]. Morphological studies of *W. ichthyophaga* have shown that changes in the size of cell aggregates and the thickness of the cell wall are crucial. The thickness of the cell wall can increase up to three-fold, while the size of cell clumps can increase even more [6].

In contrast to the large and duplicated genome of *H. werneckii*, the genome of *W. ichthyophaga* is very compact, with 9.6 Mb and only 4884 predicted protein-coding genes [49]. Genome analysis showed that except for the enriched protein family of P-type ATPases, cation transporters are sparse and exhibit low expression levels. Additionally, the expression of most of them is independent of salt concentration. Conversely, a significant expansion of hydrophobins, small amphipathic proteins that reside in the cell wall of filamentous fungi and are involved in a range of processes of cellular growth and development, was found. This significant expansion is specific for *W. ichthyophaga*, as from the estimated 15 hydrophobin genes found in the genome of a common ancestor of *W. ichthyophaga*, the gene number increased to 26 in halophilic *W. ichthyophaga* [49] Compared to those of other fungi, hydrophobins in *Wallemia* contain a higher proportion of acidic amino acids on their surface. As this is similar to archaeal halophilic proteins [50] it is reasonable to suggest that hydrophobins play a role as halophilic proteins. With their acidic amino acids exposed at the surface, they can bind water and salt and facilitate adaptation to salt exposure. Transcriptome analysis supported this idea, as >50% of hydrophobins are differentially expressed at different salinities [49]. 

*W. ichthyophaga* use the strategy of compatible solutes and, as in halotolerant *H. werneckii*, glycerol is the main compatible solute in *W. ichthyophaga*. The *W. ichthyophaga* genome contains genes for the enzymes that are involved in glycerol synthesis. The homologue of *GPD1*, *WiGPD1*, exhibited lower expression levels and a slower response to hyperosmotic shock in comparison to that of halotolerant *H. werneckii*. As in halotolerant *H. werneckii*, the expression of this gene is regulated by MAP kinase WiHog1 [51]. Genes such as *SLT1* for the plasma membrane glycerol/H+ symporter and *FPS1*, which codes for aquaglyceroporin channel Fps1, were found in multiple copies in *W. ichthyophaga,* suggesting their involvement in regulating glycerol levels. Among other proteins, the main regulatory enzyme in sterol synthesis, HMG-CoA reductase, also belongs to salt-responsive proteins in *W. ichthyophaga* and responds with a characteristic salt-dependent U-shape activity pattern [19]. The high activity and protein levels of HMG-CoA reductase in *W. ichthyophaga* under hypo- and hyper-saline conditions suggest that, as in extremely halotolerant *H. werneckii,* HMG-CoA reductase (with NaCl-dependent responses) plays an important role in the physiological adaptation of *W. ichthyophaga* to both hypo- and hyper-saline conditions.

## 5. The HOG Signal Transduction Pathway in the Halophilic Fungus *W. ichthyophaga*

As demonstrated by whole genome analysis [49], *W. ichthyophaga* contains all key genes of the HOG signal transduction pathway (Figure 2). Further molecular studies revealed similarities as well as important differences in the molecular machinery involved in sensing and responding to changes in NaCl concentrations between halophilic *W. ichthyophaga* and halotolerant *H. werneckii* and other fungi. Although many key proteins of the HOG pathway are conserved in halophilic *W. ichthyophaga*, the architecture and activation of the HOG pathway are specific. While all genes of the HOG components are present in two copies in *H. werneckii*, in *W. ichthyophaga*, all HOG pathway components, except the final MAP kinase Hog 1 (WiHog1 A and B), are present in only one isoform [51,52].

### 5.1. The Sensory Apparatus of the HOG Pathway in W. ichthyophaga

Compared to *S. cerevisiae* and halotolerant *H. werneckii*, *W. ichthyophaga* lacks a considerable number of orthologs that are involved in the osmosensing apparatus of the SHO1 branch of the HOG pathway, e.g., the mucins Msb2 and Hkr1 and the membrane anchor Opy2. Conversely, other orthologs of this branch, e.g., WiCdc24, WiCdc42, WiSte20, WiSte50, and WiSte11, are present [52]. Despite the presence of proteins of the SHO1 sensory apparatus in halophilic *W. ichthyophaga*, further studies did not confirm activation of the HOG pathway through this branch. Structural analysis of WiSho1 domains, important for interactions with downstream protein partners, showed that crucial motifs for the interactions with MAPKKK Ste11 MAKK Pbs2 are not conserved in *W. ichthyophaga* orthologs. Moreover, WiSho1 expressed in *S. cerevisiae* mutant cells did not efficiently complement the function of ScSho1 [52] 

The core phosphorelay system of the SLN1-like branch is well conserved in *W. ichthyophaga* [52]. This system contains WiYpd1, WiSsk1, the kinase WiSsk2, and the response regulator WiSkn7. Conversely, no membrane-bound ScSln1 ortholog was found in *W. ichthyophaga*. Instead, as in the genome of halotolerant *H. werneckii*, we confirmed the Group III cytosolic histidine kinase WiNik1. The *W. ichthyophaga* genome contains four hybrid histidine kinases that belong to groups I or II, III, and VIII. It was demonstrated for several fungi that group III histidine kinases are involved in two-component HOG pathway signaling [30]. We proposed that the HAMP domain repeats (present in histidine kinases, adenyl cyclases, methyl-accepting proteins, and phosphatases) [53] in WiNik1 revealed by structural studies [52] might be involved in osmosensing in *W. ichthyophaga*, as has been demonstrated for DhNik1 in moderately halotolerant *D.*
*hansenii*, CaNik1 in *C.*
*albicans* [54] and ClNik1 in *Candida lusitaniae* [39] It has been shown that group III histidine kinases are sensitive to the fungicide fludioxonil, and thus this fungicide represents a potent tool to investigate group III histidine kinase signaling [55]. Our investigation of the sensitivity of *W. ichthyophaga* cells to fludioxonil demonstrated that the presence of fludioxonil completely inhibited the growth of *W. ichthyophaga* cells at all NaCl concentrations used. This result strongly implies that WiNik1 plays a role in osmosensing machinery.

### 5.2. The MAP Kinase Module of the HOG Pathway in W. ichthyophaga

As MAPKK Pbs2 is the central element of the HOG pathway that can in principle receive hyperosmolarity signals through both the SHO1 and SLN1 branches [56], we further analyzed WiPbS2, which was identified in the *W. ichthyophaga* genome. Structural domains, important for interactions with upstream proteins involved in WiPbs2 signal transmission, were compared with ScPbs2 and the orthologs of some other fungi (27) It was demonstrated that the proline-rich motif that is crucial for Sho1 binding and signaling through the SHO1 branch in *S. cerevisiae* [57] is not conserved in *W. ichthyophaga* [52]. Another important feature of Pbs2 from *S. cerevisiae* is the docking site for MAPKKK Ste11 [25,58] Again, in WiPbs2, the Ste11 docking site is only poorly conserved [52]. Poor interactions of WiPbs2 with WiSho1 and WiSte11 indicate that WiPbs2 is not activated via the SHO1 branch, which is in line with the observation that WiSho1 could not complement the function of ScSho1 in the *S. cerevisiae* mutant strain. Conversely, we found that the ScSsk2/ScSsk22 activation of WiPbs2 is fully conserved [52] This finding and the sensitivity of *W. ichthyophaga* cells to fludioxonil highlight the importance of two-component HOG pathway signaling in *W. ichthyophaga.*

The *W. ichthyophaga* genome has two gene copies of MAP kinase Hog1, *WiHOG A*, and *WiHOG B.* Although protein sequence alignment revealed high conservation of the Hog1 kinase domains and motifs in both of the WiHog1A and WiHog1B isoforms, WiHog1A is not fully functional when compared with WiHog1B and ScHog1 in the *S. cerevisiae* mutant strain [51,52] Supported also by the observed lower phosphorylation level of WiHog1 A upon hyperosmotic stress, it was proposed that WiHog1A cannot optimally interact with the activating MAPKK Pbs2 or its targets. This might be explained by small differences in the ATP-binding site, activation loop, common docking domain, and/or PBD-2 region. Conversely, it was found that WiHog1B was fully phosphorylated; in the *S. cerevisiae hog1Δ* background, WiHog1B serves as a fully functional kinase. Moreover, WiHog1B also improves the tolerance of the yeast strain to high salinity, which was not observed for HwHog1 expression from the halotolerant *H. werneckii* in *S. cerevisiae* [13] Additionally, MAPK WiHog1 upregulated *GPD1* transcription, which is important for the synthesis of the main compatible solute, glycerol, in *W. ichthyophaga*. Again, efficient interaction of the *GPD1* promoter was demonstrated with WiHog1B but not WiHog1A [26,30].

As in halotolerant *H. werneckii*, WiHog1 transcript levels are salt-dependent [26]. Interestingly, *W. ichthyophaga* shows the opposite to the usual phosphorylation pattern found in *S. cerevisiae*, *H. werneckii*, and most other studied fungi. In *W. ichthyophaga*, MAP kinase WiHog1 is constitutively phosphorylated under optimal osmotic conditions (at concentrations of 15–20% NaCl) and is dephosphorylated under both hypo- and hyper-saline conditions. A similar phosphorylation model has so far been reported for pathogenic *C. neoformans*, serotype A, and it was assumed that a unique mechanism of Hog1 phosphorylation in *C. neoformans* is responsible for its higher stress resistance and the virulence of its serotype A [1]. Thus, it is reasonable to speculate that, in a similar manner, such HOG pathway regulation might cause the high salt resistance of *W. ichthyophaga*. This phosphorylation pattern also implies that phosphatases play an important role in HOG pathway regulation in *W. ichthyophaga*. The HOG pathway in *S. cerevisiae* is regulated by negative feedback through the action of phosphothreonine and phosphotyrosine phosphatases (17,18). In the *W. ichthyophaga* genome, we identified the phosphothreonine phosphatase WiPtc1, which is most similar to ScPtc1 and WiPtc3, which is related to ScPtc3 and its paralog ScPtc2 [20,54]. Furthermore, the *W. ichthyophaga* genome contains two phosphotyrosine phosphatases: WiPtp1 (an ortholog of ScPtp1) and WiPtp3, the latter being more related to ScPtp3, a broad-range phosphatase in *S. cerevisiae* [56] The presence of these phosphatases in the *W. ichthyophaga* genome support the importance of WiHog1 dephosphorylation in regulating the HOG signal transduction pathway.

Together, our investigation of the architecture of the *W. ichthyophaga* HOG pathway shows that the interactions of WiPbs2 kinase and the SHO1-branch orthologs are not conserved, which in turn suggests that these orthologs are not involved in WiPbs2 activation. Conversely, the ScSsk2/ScSsk22 activation of WiPbs2 is fully conserved, and *W. ichthyophaga* cells are sensitive to fludioxonil, which highlights the importance of the two-component HOG pathway signaling in *W. ichthyophaga* [39]. A unique WiHog1 phosphorylation pattern was demonstrated to be necessary for the high (er) stress-resistance of *C. neoformans* [1] and could also help *W. ichthyophaga* to survive in its naturally hypersaline environments.

In this review, the importance of the HOG pathway in adapting to higher NaCl concentrations in halotolerant *H. werneckii* and the obligate halophile *W. ichthyophaga* is summarized. Although there are many similarities between halotolerant and halophilic fungi, molecular analyses of whole genome sequences of both fungi performed in our laboratory during the past years have revealed important differences in sensing and responding to severe changes in NaCl concentrations, typical for the natural ecological niche of *H. werneckii* and *W. ichthyophaga*. We confirmed that the HOG pathway plays an important role in adaptation in both fungi. Studies revealed that the MAP kinase module is conserved in both fungi; however, the mechanisms of activation differ between halotolerant *H. werneckii* and halophilic *W. ichthyophaga*. In *H. werneckii*, the HOG pathway is activated via both SHO and SLN branches, whereas in *W. ichthyophaga*, only the SLN branch appears to be involved. Secondly, HwHog1 is phosphorylated in halotolerant *H. werneckii* and dephosphorylated in halophilic *W. ichthyophaga* under hypersaline conditions. Molecular investigations of extremely halotolerant *H. werneckii* and halophilic *W. ichthyophaga*, isolated from the same hypersaline environments, demonstrate that different fungi can employ diverse mechanisms to combat the same harsh conditions. We can also speculate that these differences make an important contribution to the halotolerant and halophilic characteristics of *H. werneckii* and *W.ichthyophaga*.

## Figures and Tables

**Figure 1 jof-07-00988-f001:**
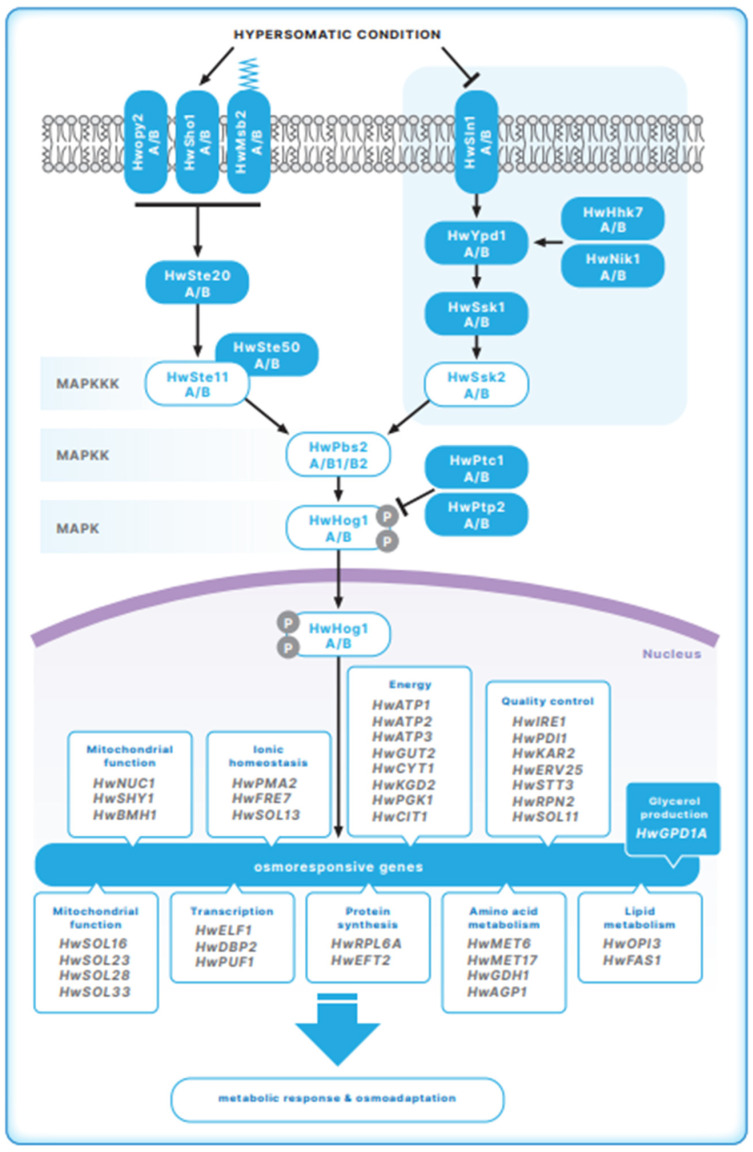
Model of the HOG pathway architecture in *H. werneckii***.** Putative HOG-pathway components identified in the genome of *H. werneckii* are shown. HwSho1 and kinases Hw HhK, HwSte11, HwPbs2 and HwHog1 proteins were further characterized as summarized in the text. Target osmoresponsive genes of HwHog1 identified in *H. werneckii* in our studies [37], are also presented (Illustration made by Matej Kocjan).

**Figure 2 jof-07-00988-f002:**
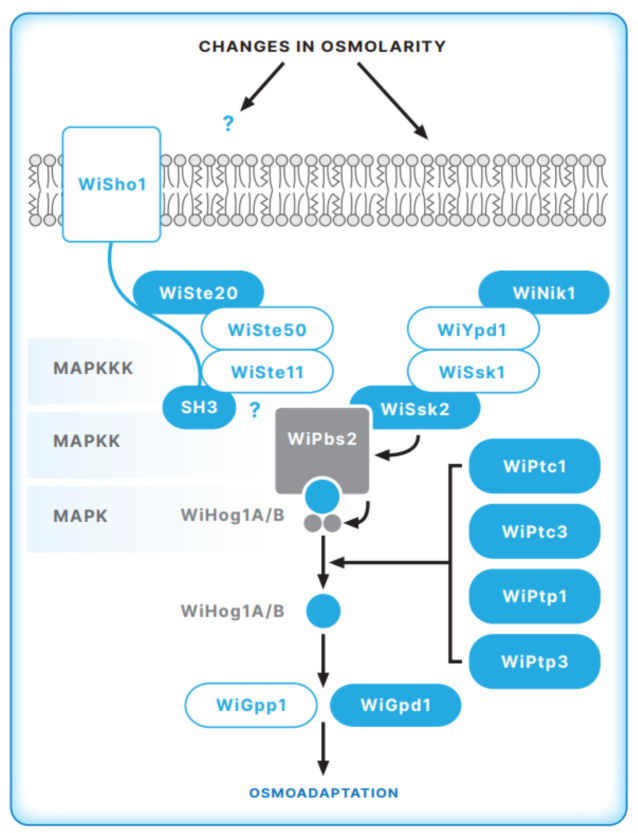
Model of the HOG pathway architecture in *W. ichthyophaga***.** Putative HOG-pathway components identified in the genome of *W. ichthyophaga* are shown. WiSho1, WiSte11, WiPbs2, and WiHog1 proteins were characterized in detail as summarized in the text. Identified target genes of WiHog1, WiGpp1, and WiGpd1are shown. (Illustration made by Matej Kocjan).

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
