# Peer review of "Sensing and Responding to Hypersaline Conditions and the HOG Signal Transduction Pathway in Fungi Isolated from Hypersaline Environments: Hortaea werneckii and Wallemia ichthyophaga"

_jof, 2021, doi:10.3390/jof7110988_

Round 1

Reviewer 1 Report

The manuscript entitled: Sensing and responding to hypersaline conditions and the HOG signal transduction pathway in fungi isolated from hypersaline environments: Hortaea werneckii and Wallemia ichthyophagais a revision focused of the structure and function of the HOG signal transduction pathway in two fungi, a salt- tolerant, Hortaea werneckii and a halopjilic fungus Wallemia ichthyophaga. Author demonstrates knowledge and expertise in the field since most of the experimental articles that appear in the bibliography belong to the author.

I find the review interesting although the researchers interested in these fungi can be scarce. The manuscript is well structured although I find some mistakes that should be solved before the publication of the manuscript.

Line 148 “and phosphotyrosine phosphatases” please erased the underlined and metches sizes

Revise of the references , specially paragraph form line 150 to 161, many of the references that appear in the test are no present in the bibliography, eg; Ma and Li, 2013 and de Oliveira Bruder Nascimento 2016. Line 184 Lenassi 2013 and Lenassi 2006. These two references appear as Lenassi et all , I guess author means et al. The format of the references is not standardized: eg. Hohmann et al., 2007, Hohmann 2009…(line 142).  I do not think that Prieto 2014 is the best reference to illustrate the histidine kinase in Candida albicans (line 162).

Please improve the quality of figure 1 and revise the figure caption. Putative should not be written in italic, and the following sentence “Shown are also identified Hog1 target osmoresponsive genes”line 175 is not well structured. Have the blue and white color any special meaning? Please, explain.

Please, revise the name of the genes line 204 HwHH7A, should be HwHHK7A, the same in line 208 and 214.

Line 236-238  The sentence “the SLN1 branch, which is similar to that in pathogenic fungi such as A. nidulans and 236 C. albicans in which histidine kinases were proposed to be responsible for transmitting the 237 signal to the YpdA-SskA-SskB phosphorelay” is not correct since YpdA-SskA-SskB belong to Aspergillus not to Candida.

Line 260 “and the effect of the Hog1 kinase inhibitor” Which Hog1 kinase inhibitor does author refer to?

Line 273 30% M NaCl, I guess that M should be erased

Line 279-280 “the HwHog1 phosphorylation signal are involved in the effects of nuclear HwHog1 in H. werneckii” Why nuclear effects?

Revise the name of the genes, line 341 STL1  and FPS1 are not in italics.

Figure2 line 366  WiGpd1 is not properly written.

I don´t understand the meaning of the following sentence “The core phosphorelay system of the SLN1-like branch is well conserved in ScSho1 382 in W. ichthyophaga” Line 382. Please, revise.

Line 459, please correct  high(er.

Author Response

Revise of the references , specially paragraph form line 150 to 161, many of the references that appear in the test are no present in the bibliography, eg; Ma and Li, 2013 and de Oliveira Bruder Nascimento 2016. Line 184 Lenassi 2013 and Lenassi 2006. These two references appear as Lenassi et all , I guess author means et al. The format of the references is not standardized: eg. Hohmann et al., 2007, Hohmann 2009 (line 142).  I do not think that Prieto 2014 is the best reference to illustrate the histidine kinase in Candida albicans (line 162).

  • Reference Ma and Li 2013 is added to the list of references and instead of de Oliveira Bruder Nascimento, reference of Furukawa 2005 from the list of references is added as reference here..
  • Lenassi 2006 is, in accordance with the list of references, corrected to Lenassi and Plemenitaš 2007.
  • Homan et al.2007 is, in accordance with the list of references, in the text replaced by Homan 2002.The format of the references is standardized
  • Reference Prieto 2014 is replaced by Yamado Okabe et al.1999 and Raman et al. 2020. Both references are added to the list of references.

Please improve the quality of figure 1 and revise the figure caption. Putative should not be written in italic, and the following sentence “Shown are also identified Hog1 target osmoresponsive genes”line 175 is not well structured. Have the blue and white color any special meaning? Please, explain.

  • In Fig 1, HYPERSOMATIC CONDITION is corrected to HYPEROSMOTIC CONDITION AND the font size of metabolic response & osmoadaptation   in Fig1 is increased.
  • Putative is corrected in the Fig.1 caption
  • sentence “Shown are also identified Hog1 target osmoresponsive genes is corrected to: Target osmoresponsive genes of HwHog1 identified in H.werneckii in our studies (Vaupotič and Plemenitaš, 2007b), are also presented.
  • Blue and white colour do not have special meaning and are used by illustrator of  1 and 2 more for the artistic impression

Please, revise the name of the genes line 204 HwHH7A, should be HwHHK7A, the same in line 208 and 214.

Names of the genes are corrected to HwHHK7A

Line 236-238  The sentence the 237 signal to the YpdA-SskA-SskB phosphorelay” is not correct since YpdA-SskA-SskB belong to Aspergillus not to Candida.

The sentence is corrected to:  “the SLN1 branch, which is similar to that in pathogenic fungi such as A. nidulans and  C. albicans in which histidine kinases were proposed to be responsible for transmitting the signal to the MAP kinase module of HOG pathway.

Line 260 “and the effect of the Hog1 kinase inhibitor” Which Hog1 kinase inhibitor does author refer to? To inhibitors BPTIP and SB203580 as described in ref.Kejžar et al.2015

Line 273 30% M NaCl, I guess that M should be erased. M is erased

Line 279-280 “the HwHog1 phosphorylation signal are involved in the effects of nuclear HwHog1 in H. werneckii” Why nuclear effects?    Nuclear is erased

Revise the name of the genes, line 341 STL1  and FPS1 are not in italics. Names are corrected to italics

Figure2 line 366 WiGpd1 is not properly written. Corrected in Fig.2 caption

I don´t understand the meaning of the following sentence “The core phosphorelay system of the SLN1-like branch is well conserved in ScSho1 382 in W. ichthyophaga” Line 382. Please, revise.

Sentence is corrected to: The core phosphorelay system of the SLN1-like branch is well conserved in W. ichthyophaga” 

Line 459, please correct  high(er. Corrected in the text

Reviewer 2 Report

This review is truly excellent. I knew little about this subject before reading it, and afterwards I feel highly informed. It will be my go-to resource for information on this topic. The writing is excellent and the organization of the review is very logical and clear. I have only the most minor of comments.

Minor comments:

Lines 96-98, “as maintenance of appropriate K+/Na+ ratios is not crucial only at high NaCl concentrations but also at low NaCl concentrations”: I think it would be clearer to move the word “not” to between “crucial” and “only”.

Lines 109-110: “HMG-CoA reductase can be considered an important bio-chemical signature of halophilism”: to me, this statement implies that the PRESENCE of HMG-CoA reductase is a signature of halophilism, but this enzyme is also present in non-halophilic organisms (like S. cerevisiae). Perhaps the author means to say that high or regulated expression of HMG-CoA reductase can be considered a signature?

Line 148: unexpected formatting (underline) for “ ) and phosphotyrosine phosphatases”.

Figure 1: the uppermost text says, “Hypersomatic”, presumably should be “hyperosmotic”. I also recommend increasing the font size of the lowermost text (“metabolic response and osmoadaptation”).

Line 389: “HAMP domain repeats”: does the acronym “HAMP” need to be spelled out? I am not familiar with these domains.

Reviewer identity: Michael McMurray, PhD University of Colorado Anschutz Medical Campus

Author Response

Lines 96-98, “as maintenance of appropriate K+/Na+ ratios is not crucial only at high NaCl concentrations but also at low NaCl concentrations”: I think it would be clearer to move the word “not” to between “crucial” and “only”

The word “not” is moved between “crucial” and “only”

Lines 109-110: “HMG-CoA reductase can be considered an important bio-chemical signature of halophilism”: to me, this statement implies that the PRESENCE of HMG-CoA reductase is a signature of halophilism, but this enzyme is also present in non-halophilic organisms (like S. cerevisiae). Perhaps the author means to say that high or regulated expression of HMG-CoA reductase can be considered a signature?

Corrected to: regulated expression and activity of HMG-CoA reductase

 Line 148: unexpected formatting (underline) for “ ) and phosphotyrosine phosphatases”. Corrected

 Figure 1: the uppermost text says, “Hypersomatic”, presumably should be “hyperosmotic”. I also recommend increasing the font size of the lowermost text (“metabolic response and osmoadaptation”).

Thank you so much for this remark.  Hypersomatic” is corrected “hyperosmotic” and the font size of the lowermost text is increased

 Line 389: “HAMP domain repeats”: does the acronym “HAMP” need to be spelled out? I am not familiar with these domains

 Following  HAMP domain repeats, (present in Histidine kinases, Adenyl cyclases, Methyl-accepting proteins and Phosphatases) I added in this sentence
